# Wilderness Is the Prototype of Nature Regardless of the Individual’s Connection to Nature. An Empirical Verification of the Solastalgia Effect

**DOI:** 10.3390/ijerph20146354

**Published:** 2023-07-13

**Authors:** Giuseppe Barbiero, Rita Berto, Giulio Senes, Natalia Fumagalli

**Affiliations:** 1GREEN LEAF—Laboratory of Affective Ecology, University of the Valle d’Aosta, 11100 Aosta, Italy; rita.berto@univda.it; 2Department of Agricultural and Environmental Sciences, University of Milan, 20133 Milan, Italy; giulio.senes@unimi.it (G.S.); natalia.fumagalli@unimi.it (N.F.)

**Keywords:** connection to Nature, perceived restorativeness, domesticated Nature, urban Nature, wilderness, solastalgia

## Abstract

(1) Background: Connectedness with Nature is a personality trait that influences our relationship with Nature. But Nature is not all the same. Wilderness is Nature in its original form, the form within which human beings have evolved as a species, while what we refer to as domesticated and urban Nature are relatively recent products of our interaction with the environment. (2) Aim: The main purpose of this study was to verify whether the individual trait “connection to Nature” influences the perception of restoration, preference for and familiarity with three types of Nature: wilderness, domesticated and urban. (3) Results: Regardless of the level of connection to Nature, wilderness is always perceived as more restorative than the domesticated or urban environment. Individuals with higher connectedness prefer wilderness more than others, and they are able to recognise the restorative value of domesticated environments more than those with medium or low levels of connectedness. Less connected individuals tend to prefer domesticated environments, although wilderness is more familiar to them. (4) Conclusions: This study shows that, despite our detachment from Nature, wilderness is the prototype of Nature, and this finding offers a plausible evolutionary explanation of solastalgia.

## 1. Introduction

Literature has shown that the subjective feeling of connection to Nature (In this paper, “Nature” is written with the capital “N” to indicate the biosphere and the abiotic matrices (soil, air, and water) where it flourishes, and to avoid confusion with “nature” as the intrinsic quality of a certain creature and/or phenomenon) is associated with more positive evaluations of natural settings [1]. Evidence also shows that the sense of connection to Nature is a characteristic of the individual that does not depend on the characteristic of the environment to which he/she is being exposed; instead, the perceived restorativeness to an environment depends entirely on the specific characteristics of that environment [2]. Moreover, although the literature reports connection to Nature and the perceived restorativeness of Nature to be independent phenomena, the study by Berto et al. [3], in addition to confirming the relationship between connection to Nature and preference for Nature, shows that the biophilic quality of the environment [4] can be associated with the individual’s level of connection to Nature and perceived restoration. Therefore, preference for Nature, the perceived restorativeness of Nature restoration and connection to Nature seem to be intertwined and associated with the biophilic quality of the environment, which can be roughly summarised as the environment’s naturalness plus its functional and aesthetic value. The biophilic quality of an environment is determined by the characteristics which makes it “objectively” according to the set of adaptation rules evolved in human beings [5]. Indeed, the perceived restorativeness of an environment is expected, to some extent, to match the biophlic quality of that environment; however, this relationship may vary according to an individual’s own sense of connection to Nature, a property which appears to develop the more a person engages with and interacts with natural environments they deem to be both aesthetically and functionally pleasing [4].

### 1.1. Three Types of Nature: Wilderness, Domesticated and Urban

The type of Nature with which an individual comes into contact can influence his/her positive (biophilic) or negative (biophobic) emotional reaction [4]. There are three basic types of Nature with which individuals can come into contact: wilderness, domesticated and urban [6]. Basically, what differentiates one type of Nature from another is the level of intensity of human intervention.

Wilderness refers to natural areas that have been only minimally impacted by human activity. These areas can include forests, mountains, deserts, and oceans. Wilderness is characterised by natural beauty, biodiversity, and ecological processes. Therefore, wilderness is important for its ecological, recreational, and spiritual value [7,8]. Domesticated environments refer to areas that have been significantly altered by human activity. These areas can include urban parks, gardens, and agricultural lands. Domesticated environments are characterised by their human-made features, and these areas are often created with a specific purpose in mind, such as providing food, recreation, or aesthetic beauty [9]. Thus, an urban environment can also be considered as an extreme case of the domesticated environment. To optimise safety and the production and distribution processes of resources, human activities can alter the environment to the point of erasing Nature. Some urban environments are modified to such an extent that they can appear totally artificial. Indeed, visible Nature has almost, although not completely, disappeared within urban environments [10].

The level of human influence is one of the main differences between wilderness and domesticated Nature. In wilderness, human influence is minimal, whereas it is significant in domesticated environments. As a result, wilderness tends to be characterised by greater levels of biodiversity and ecological processes that are more intact, while biodiversity in domesticated environments is more likely to be limited and the ecological processes altered in order to serve human needs [8]. Therefore, our experience of wilderness will differ from that of domesticated environments, being more immersive and profound [11] compared with a more controlled and structured experience in a domesticated environment.

### 1.2. The Origin of Domesticated and Urban Environments

From the evolutionary standpoint, wilderness is the protype of Nature. For about 95% of our evolutionary history, corresponding to the Middle and Upper Paleolithic Eras, we humans lived and survived in the wilderness as hunters-gatherers. It is believed that domesticated and urban environments are two fundamental environmental models that derive from two distinct moments of “rupture” from the wilderness that occurred during the history of our species. The first moment of rupture was the invention of agriculture and animal breeding, which occurred about 14,000 years ago [12]. Humans began to distinguish domestic from wild Nature. This started in the Neolithic Era, an epoch which covers approximately 5% of humanity’s evolutionary history. The second moment of rupture encompassed the Industrial Revolution. From the second half of the eighteenth-century, humans began to create urban environments, characterised by an increase in population density and a decrease in green spaces [13].

### 1.3. Perceived Restoration in Different Types of Nature

Nature’s restorativeness refers to the ability of natural environments to promote recovery from physiological stress and mental fatigue in individuals. However, little is known about the individual’s perception of restoration in relation to the three basic types of Nature, namely wilderness, domesticated and urban. Recently we investigated whether perceived restoration of different types of Nature might depend on the level of that individual’s connection to Nature [14]. In fact, individuals seem to be more attracted to the type of Nature that triggers them to feel a connection with Nature [15,16,17]. In particular, individuals with lower connection to Nature tend to prefer domesticated Nature [18,19,20]. Accordingly, we hypothesised that an individual’s perception of the restorative benefits of Nature would be affected by preference for the three basic types of Nature, and that this preference might be affected by his/her connection to Nature. Thus, a preference for wilderness, the best place in which to obtain restorative benefits, may result from a greater connection with Nature. Indeed, evidence exists suggesting that wilderness has left a deep impression within the human psyche (see for example [21]).

Table 1 presents our working hypothesis: individuals prefer the kind of Nature that matches their feeling of connection to Nature; an individual with high connection would prefer wilderness, whereas an individual with a lower connection would tend to prefer more domesticated or urban environments. An individual’s connection with wilderness could thus be considered a “personality trait” with a pleiotropic effect on their preference and perception of a restorative environment [14]. This evolutionary explanation can also account for the different experiences of Nature depending on environmental typology (wilderness, domesticated, and urban), which in turn reflects the typology of human cultural experience with Nature (Paleolithic, Neolithic, and urbanised).

### 1.4. Connection to Nature, Naturalistic Intelligence and Solastalgia

The personality trait “connection to Nature” refers to the sense of affiliation with the natural world [22], which accordingly influences each individual’s relationship with Nature. This connection can be expressed in many different ways, including physical [23], psychological [24] and spiritual [25] manifestations.

Since connection to Nature is related to naturalistic intelligence [26], namely the ability “to recognize flora and fauna, to make other consequential distinctions in the natural world, and to use this skill productively” [27], we hypothesised that naturalistic intelligence might originate from a high connection with Nature which arises from frequent and direct experiences of the natural world, and which is also the key way to foster and maintain individuals’ connection to Nature [28]. However, the constant rise of urbanisation in the modern world is reducing the opportunity for exposure to natural environments and favouring disconnection from Nature. Disconnection from Nature may also turn into solastalgia, a form of psychological and emotional distress caused by environmental change—in particular, the degradation of one’s home environment [29]. Solastalgia is a relatively new concept, first described in the early 2000s, characterised by feelings of loss and helplessness, as well as physical symptoms such as headaches, fatigue, and respiratory problems and nostalgia for the way things used to be [30].

### 1.5. The Aim of This Study

The discussion over what features and/or type of Nature influence an individual’s ability to appreciate Nature’s restorative benefits and their attitudes towards this phenomenon is a relatively recent addition to the research field on restorative environments. Our understanding of the relationship between exposure to Nature and psychological well-being has often been investigated by considering citizens’ proximity to green areas, a neighbourhood’s “greenness”, or the normalised difference vegetation index (NDVI, [31]), the Recreation Opportunity Spectrum (ROS, [3,32]), or from the perspective of the ecosystem, by quantifying habitat and species diversity, or ecological functions [33,34].

In addition to research on access to natural spaces in general, there is a considerable body of work that has addressed the specific aspects of Nature that confer benefits to individuals. It has long been understood that the experience of biodiversity and other aspects of the natural world can act through psychological and physiological mechanisms to enhance well-being [35,36]. However, research that has attempted to investigate the link between biodiversity in green spaces and psychological well-being has produced mixed results [33,37]. While laboratory research has suggested there is little difference between the restorative benefits of very different types of natural scenes [38], a strong cross-cultural preference exists for semi-natural green spaces as opposed to more formal parks [39]. Nevertheless, the literature contends that the quality of Nature to which one is exposed is as important for an individual’s well-being as the quantity of exposure. Evidence also sustains that focusing only on quantity is not sufficient to inform public health policy, or indeed to aid biodiversity conservation [37,40].

This study bypasses the problem of the quantity and quality of exposure to Nature because it addresses the issue from the evolutionary perspective and proposes a classification of Nature which satisfies both requirements at the same time. In each human being three fundamental phylogenetic experiences are probably present: the wilderness of the Paleolithic, the domesticated Nature of the Neolithic, and the absent Nature of the urban environment. Each of the three phylogenetic experiences of Nature create a different type of relationship between the individual and Nature, with can manifest at different intensities and vary from individual to individual [6].

This study aims to verify whether the individual’s level of connection to Nature (operationalised as low, medium and high) affects the perception of perceived restoration and the preference for wild, domesticated or urban Nature; individuals with high connection to Nature are expected to show greater preference for wilderness and be more able to perceive the restorative benefits associated with exposure to this form of Nature compared with that obtained from exposure to domesticated and urban Nature.

In light of the relationship between connection to Nature and naturalistic intelligence, in this study naturalistic intelligence was assessed as a concurrent measure of the construct “connection to Nature”; these two dimensions are expected to go in the same direction. Although literature shows that a personality trait like connection to Nature can affect the environment’s perceived restoration [14,15,16], there are no research studies considering the relationship between personality traits, connection to Nature and perceived restoration except for Takayama, Morikawa and Bielinis’ study [41]. In that study, the effect of the most common personality trait was verified on perceived restoration, but no significant result emerged. For this reason, in this study we also verified whether a relationship exists between two personality dimensions that open the individual to the natural world (i.e., connection to Nature and naturalist intelligence) and a dimension which, on the contrary, opens the individual to the other people, like interpersonal intelligence. Since literature suggests that connection to Nature is positively associated with subjective happiness, perceived satisfaction with meaningful relationships, e.g., with family members and friends, and altruism [17,42,43,44,45], in this study interpersonal intelligence was measured in order to verify if it might affect the environment’s perceived restoration and its relationship with individual’s connection to Nature and accordingly with naturalistic intelligence.

## 2. Materials and Methods

### 2.1. Participants

A total of 415 subjects were approached using a university mailing list and personal networks. 354 subjects accepted to participate in the research study; underage persons and uncompleted questionnaires were automatically excluded from the dataset. The final sample was made up of 316 subjects (215 females and 101 males), mean age = 41 years (range 20–60).

### 2.2. The Instrument

The instrument was made up of the Interpersonal and Environmental subscales from the Multiple Intelligence Profiling Questionnaire III, the Connection to Nature Scale, and the Perceived Restorativeness Scale. A description of each scale is provided below.

#### 2.2.1. Interpersonal and Naturalistic Intelligence

The Multiple Intelligence Profiling Questionnaire III (MPIQ III; [46]) measures the nine dimensions of Gardner’s Multiple Intelligence theory, which are the following: (1) Linguistic (2) Logical-mathematical (3) Musical (4) Spatial (5) Bodily-kinaesthetic (6) Interpersonal (7) Intrapersonal (8) Spiritual and (9) Environmental intelligence. For the purpose of this study, dimension (6) and (9), measuring interpersonal and environmental intelligence, respectively, were used. The label “naturalistic intelligence” was used instead of “environmental intelligence” sticking to the original terms of Gardner’s Multiple Intelligence Theory and in line with the context of this paper (Howard Gardner [26] defined naturalist intelligence as the ability “to recognize flora and fauna, to make other consequential distinctions in the natural world, and to use this skill productively”. Tirri and Nokelainen [45] operationalised naturalistic intelligence just as it was defined by Gardner, but they named it “environmental intelligence”, without providing an explanation for this change. Unfortunately, “environmental” is an ambiguous adjective because it can refer to both natural and artificial environments. For the purposes of this article, the items of the environmental intelligence scale will be attributed to naturalistic intelligence).

The items of the two subscales are as follows.
Interpersonal Intelligence subscale
*Even in strange company, I easily find someone to talk to.**I get alone easily with different types of people.**I make contact easily with other people.**In negotiations and group work, I am able to support the group to find a consensus.*Naturalistic Intelligence subscale
*I enjoy the beauty and experiences related to nature.**Protecting the nature is important to me.**I pay attention to my consumption habits in order to protect environment.*

Items were assessed on a Likert-scale from 1 (not at all) to 5 (completely). The average score of each subscale’s items establishes the measure of one’s interpersonal (INTERP) and naturalistic (NATUR) intelligence.

#### 2.2.2. Connection to Nature

To assess the feeling of affiliation with Nature, the Connection to Nature Scale (CNS) was used. This CNS is a slightly modified version of Berto, Pasini and Barbiero ([2], original version by [22]) which allows for evaluating the extent to which an individual feels part of the natural world. This version of the CNS is a reliable measurement (alpha of Cronbach = 0.91) of the “affiliation with Nature” construct of the Biophilia Hypothesis [47,48]. The CNS consists of seven items rated on a 5-point scale, where 1 = never, and 5 = always. The average score of the 7 items indicates one’s personal relationship with Nature. The CNS items follow.
*I feel connected to all animals and plants I see around me.**I feel I am a part of the same world of grass and hunts.**I think butterflies are smart too.**I feel connected to trees and birds.**I feel I belong to the sea and the sea belongs to me.**I feel like part of a meadow.**I feel part of the natural world like a tree is a part of the wood.*

#### 2.2.3. Perceived Restoration

The Perceived Restorativeness Scale-11 (PRS-11; [49], based on the original version by [50]), was used to measure the individual’s perception of four restorative factors:being-away (BA; 3 items): a setting that allows physical and/or psychological distance from demands on directed attention; items are:*Places like that are a refuge from nuisances.**To get away from things that usually demand my attention I like to go to places like that.**To stop thinking about the things that I must get done I like to go to places like that.*fascination (FA; 3 items): the type of attention stimulated by interesting objects, namely a setting that provokes curiosity in the individual and fascination about things, and is assumed to be effortless and without capacity limitations; items are:*Places like that are fascinating.**In places like that my attention is drawn to many interesting things.**In places like that it is hard to be bored.*coherence (COH; 3 items): a setting where activities and items are ordered and organised; items are:*There is a clear order in the physical arrangement of places like that.**In places like that it is easy to see how things are organised.**In places like that everything seems to have its proper place.*scope (SCO; 2 items): a setting that is large enough such that it does not restrict movement, thereby offering a sort of “world of its own”; items are:*That place is large enough to allow exploration in many directions.**In places like that there are few boundaries to limit my possibility for moving about.*

Additional items were included to the PRS-11 in order to assess familiarity (FAM; 1 item): *That place is familiar to me*, and preference (PREF; 1 item): *I like that place*.

Items are rated on a 0 to 10-point scale, where 0 = not at all, 6 = rather much, and 10 = completely. The mean perceived restorativeness (PR) score was the summary score of the items assessing the restorative factors, while the preference (PREF) and familiarity (FAM) score derived from the answer to the single items.

### 2.3. Stimulus Material

A large number of photographs of different types of Nature were systematically collected from existing stimulus material and assessed by four independent judges; the goal was to sample three types of Nature: wild, domesticated and urban, one image per category. The three selected images are shown in Figure 1.

### 2.4. Procedure

The data collection took place during the COVID-19 Pandemic; accordingly, the subject recruitment and instrument administration was online. After accepting the invitation, each participant was given a brief general overview of the study, while remaining blind to the research hypothesis, and then was directed to one version of the instrument where the scales were presented in a randomised order. The PRS-11 had to be filled in three times, one per image and the image presentation was randomised as well. The data collection was in digital form and took 3 weeks.

## 3. Results

First, the mean Connection to Nature (CN) score was calculated for all participants (M = 3.77, SD = 0.95), then participants’ CN score was sorted out in Low (CNS score < 3), Medium (CNS score > 3 and <4) and High (CNS score > 4) (Table 2). Participants showed a Medium to High level of CN.

The mean scores of participants’ interpersonal (INTERP) and naturalistic (NATUR) intelligence were calculated across the three levels of CN (Table 3). A multivariate ANOVA was run on these scores to verify the effect of CN on these two forms of Intelligence. The significant effect of CN emerged for both variables between subjects: INTERP, F(2, 313) = 10.41, *p* = 0.000; NATUR, F(2, 313) = 45.18, *p* = 0.000; *p* < 0.05. Bonferroni’s multiple comparisons showed significant differences between all levels of CN for INTERP (all *p* < 0.05) and NATUR (all *p* < 0.05). Participants’ interpersonal and naturalistic Intelligence varied significantly based on their level of connection to Nature: participants with a higher connection to Nature also showed higher interpersonal and naturalistic intelligence and vice versa.

The participants’ mean score of perceived restoration (PR), preference (PREF) and familiarity (FAM) for the three types of Nature: wild, domesticated and urban (W, D, U) was calculated across the three levels of CN (Table 4).

To verify the effect of CN on PR, PREF and FAM of the three types of Nature (W, D, U), a Repeated Measure ANOVA was run on these means (CN: fixed factor). The following significant effects turned out: the significant interactions between types of Nature * levels of CN on PR: F(4, 622) = 11.57, *p* = 0.000; on PREF: F(4, 622) = 7.52, *p* = 0.000; and on FAM: F(4, 622) = 6.84. *p* = 0.000; the significant effect within subjects of PR: F(2, 622) = 300.10, *p* = 0.000; of PREF: F(2, 622) = 208.77. *p* = 0.000; of FAM: F(2, 622) = 34.16. *p* = 0.000; the significant effect between subjects of CN on PR: F(2, 311) = 7.63, *p* = 0.001; on PREF: F(2, 311) = 4.91, *p* = 0.008; on FAM: F(2, 311) = 5.81. *p* = 0.003; (all *p* < 0.01). Participants’ level of connection to Nature significantly affected the assessments of each image. However, participants’ perceived restoration, preference and familiarity of each image also varied significantly based on the type of Nature. The graphs (Figure 2) show the interaction between types of Nature * levels of CN on the dependent variables. Perceived restoration and preference show the same trend for the three levels of connection to Nature across the three types of Nature.

It is worth noting that the average preference scores of low connected individuals for wilderness and domesticated environments were 2 points higher than the average scores attributed to the urban environment. The same trend, even if less pronounced, emerged for perceived restoration. On the other hand, individuals highly connected preferred wilderness, but they perceived domesticated environments more restorative in comparison to medium or low connected individuals.

Pearson’s correlation was calculated between INTERP, NATUR, CN, PR, PREF and FAM; these correlations were calculated on the mean scores of the entire sample with no distinction of the CN level (Table 5). Participants’ connection to Nature correlates slightly with their interpersonal intelligence, with perceived restoration, preference and familiarity of the image. On the contrary, participants’ perceived restoration correlates well with familiarity and highly with the preference for the image. A good correlation emerged between participants’ connection to Nature and naturalistic intelligence.

The Pearson’s correlation was calculated on the same variables making the distinction on the CN level and without considering the type of Nature (Table 6). Only at the low level of CN participants’ naturalistic intelligence correlates moderately with perceived restoration and preference of the image. Participants’ perceived restoration correlates with preference and familiarity of the image at all levels of connection to Nature, as well as naturalistic intelligence and connection to Nature.

Pearson’s correlation was now calculated between PR, PREF and FAM scores of the three types of Nature across the three levels of CN. This further distinction showed significant correlations ranging from slight to moderate values between variables. The most interesting correlations are negative. At the low level of CN, participants’ preference for wilderness correlates negatively with their preference for urban Nature: r = −0.26 (*p* < 0.05). At the medium level of CN, participants’ naturalistic intelligence correlates negatively with their perceived restoration for urban Nature: r = −0.24 (*p* < 0.05), and their preference for urban Nature: r = −0.22 (*p* < 0.05). No negative correlations emerged for familiarity, but at the low level of CN, familiarity for urban Nature correlates with participants’ interpersonal intelligence: r = 0.29 (*p* < 0.05).

## 4. Discussion

Nature’s restorativeness refers to the ability of natural environments to promote recovery from physiological stress and mental fatigue in individuals. In this case it is about restorative environments [51]. Nature’s restorativeness is perceived as such by individuals when the natural environment is distinct (either physically or conceptually) from everyday environments and extents in terms of time and space, is coherent, engages the mind and promotes exploration. Environments with these characteristics are the best candidates to answer the individual’s need for perceived restoration. But can this need of restoration be affected by the individual’s sense of connection with Nature? The objective of this study was exactly to verify whether individual connection with Nature might affect the restorative perception and preference for wild, domesticated and urban Nature. Results showed that wilderness as being perceived more restorative than urban Nature independently of participants’ level of connection to Nature. Preference assessments showed the same trend of perceived restoration, as also confirmed by the high correlation between these two variables. Overall, participants preferred wilderness to urban independently of their level of connection to Nature, but among participants those who had low connection to Nature preferred domesticated Nature better than wilderness. Moreover, participants with a low connection to Nature perceived domesticated Nature as being slightly more restorative than wilderness, but they perceived wilderness as more restorative than urban Nature; they also showed greater appreciation for urban Nature—in all assessments—than those with high and medium connection to Nature, but always lower than their consideration for wilderness. Participants with a high connection to Nature preferred and perceived the restorative benefits associated with wilderness better than all other participants. This result seems to sustain the role of connectedness to Nature in the process that generates awe in enhancing an individual’s well-being [52]. Furthermore, those with a high connection to Nature were also those who appreciated the restorative value of domesticated Nature the most. In brief, those with a high connection to Nature showed more appreciation for Nature whether it was wild or domesticated. This could also explain why connection to Nature correlates well with naturalistic intelligence but not as well with interpersonal intelligence. Experience with Nature seems to be a separate category of experience, even though in natural environments people tend to behave more altruistically [53].

The results show that the ability to perceive the restorative benefits of wilderness is conserved, at least to some extent, in individuals with a medium or low connection to Nature. This result is surprising, indeed, especially in the case of individuals with a low connection to Nature. In fact, although one might expect wilderness to be preferred by individuals highly connected to Nature [3,14], which one would also expect to correlate with high perceived restoration and familiarity of wild Nature, it is less clear why individuals with low levels of connection would also prefer wilderness as well. Instead, one might expect these individuals to prefer urban environments the most (see Table 1; [18,19]). Although individuals with a low connection to Nature were shown to prefer domesticated Nature slightly more than wilderness, they nevertheless perceived wilderness as being more restorative than urban environments. One might have expected this result to depend on individuals with a low connection to Nature having low familiarity of the wilderness, but this was not the case. Paradoxically, individuals with low connection to Nature reported wilderness as being more familiar to them than urban Nature, which was more familiar to them than domesticated Nature (being the least familiar of all three types), thus, overturning, to some extent, the assessments of preference, i.e., in these individuals, the trends in preference, perceived restoration and familiarity were misaligned (Figure 2).

We can attempt to explain this apparent contradiction by considering the adaptations that our species has developed during its evolution in wilderness. The sensory apparatus and the consequent psychological adaptations evolved to respond to the selective pressures exerted by the wilderness. Our results show that the psychological adaptations of biophilia described by E.O. Wilson [54]—fascination for Nature (of which perceived restoration being its measure, [14]) and affiliation with Nature—remain intact, regardless of personal experience which may lead to a greater or lesser connection with Nature.

These results can be considered the first empirical observation substantiating the solastalgia effect within the conceptual framework of evolutionary psychology. Solastalgia is defined as a form of psychological and emotional distress caused by environmental change, in particular the degradation of one’s home environment [29]. Therefore, if even individuals with a low connection to Nature feel more familiar with the wilderness, then the loss of wilderness could be perceived as a loss also in terms of restorative possibilities, and the feeling of affiliation with Nature (one of the two constructs of the biophilia hypothesis [47,54]) would also be expected to be affected.

Solastalgia can be described by two models of preference: the model proposed by Whitfield [55] and the discrepancy model by Purcell [56]. Both models are concentric in that they focus on the prototype, which the authors place at the centre of the preference evaluation process, and the set consists of all environmental instances. The models differ in the location of the individual’s maximum preference, which coincides with the prototype in Whitfield’s model, whereas in Purcell’s model it corresponds to a position halfway between the prototype and the external margin of the set (Figure 3). If, in the light of our results, we place wilderness as the prototype in Figure 3 and consider preference for the three types of Nature along with the individuals’ level of connection to Nature, we can hypothesise that Whitfield’s model would work better for individuals with a high connection to Nature, while Purcell’s model would be more accurate for those with a low connection to Nature. Accordingly, perceived restoration depends not only on the individual’s connection to Nature, but also on how far the preferred natural environment is from the wilderness prototype. These models allow us to introduce a cognitive explanation of solastalgia. Individuals with a high connection to Nature are expected to experience solastalgia when they are in an urban environment that is far from the prototype of wilderness (Whitfield’s model), whereas individuals with a low connection to Nature should suffer from solastalgia when they move away from the prototype of wilderness. This could explain their (alleged) familiarity with wilderness, whilst preferring and perceiving domesticated Nature (which is positioned in an intermediate position between wilderness and urban environment) as more restorative than urban Nature (Purcell’s model). Therefore, we can hypothesise for individuals to overcome biophobia, which mainly occurs in individuals with a low connection to Nature [57], the best strategy would be for them to make initial contact with domesticated Nature in protected environments (perceived as more restorative than wilderness) before they approach wilderness [58] (pp. 185–207), the more familiar ancestral prototype.

## 5. Conclusions

An evolutionary bias is present in humans which guides them to exhibit a preference for certain kinds of environments [60], namely those with attributes resembling the natural environments in which humans evolved [61,62]. Environmental preference can be considered the expression of an adaptative behaviour that brings the individual to distance him/herself from inappropriate environments and direct him/her towards a more desirable environmental setting. This innate preference for natural (wilderness and domesticated) environments over artificial/urban ones constitutes an adaptation in our evolution process [63]. Environmental preference is affected by an individual’s need to be restored [64], and the general preference for natural environments can be explained by the notion that “psycho-physiological” restoration occurs more easily in natural environments; people tend to prefer natural environments because they allow an individual’s psycho-physiological well-being to be maintained or even enhanced [65].

This study shows that wilderness is, and remains, in the psyche of human beings, the prototype of Nature: wilderness is the original environment in which our species evolved. The results presented here also offer an explanation of solastalgia consistent with the evolutionary perspective, specifically that humans’ adaptations are the result of being accustomed to a given environment for a very long time, and that such adaptations cannot be changed within a few generations.

## Figures and Tables

**Figure 1 ijerph-20-06354-f001:**
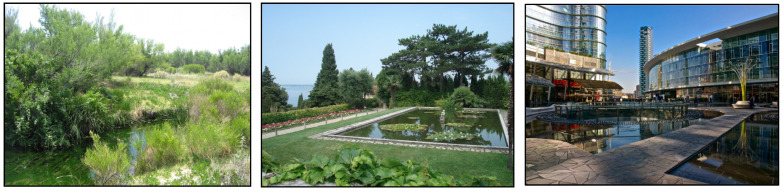
The three images of Nature selected. From the left: wild (W), domesticated (D) and urban (U) Nature.

**Figure 2 ijerph-20-06354-f002:**
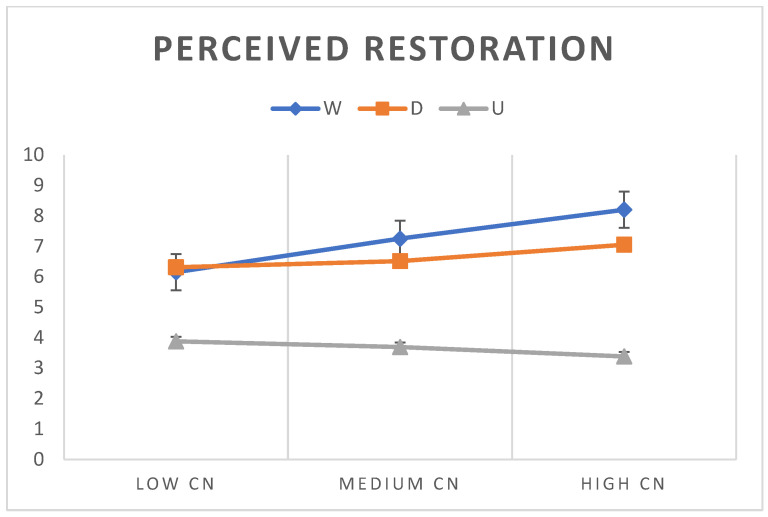
Graphic representation of the results of Table 4. Urban Nature always obtains clearly lower assessments in perceived restoration (above) and in preference (in the centre), regardless of the connection to Nature. Conversely, wilderness is perceived as the most restorative environment and preferred by individuals with medium and high connection to Nature, while individuals with lower connection to Nature perceive domesticated Nature as the most restorative and preferred.

**Figure 3 ijerph-20-06354-f003:**
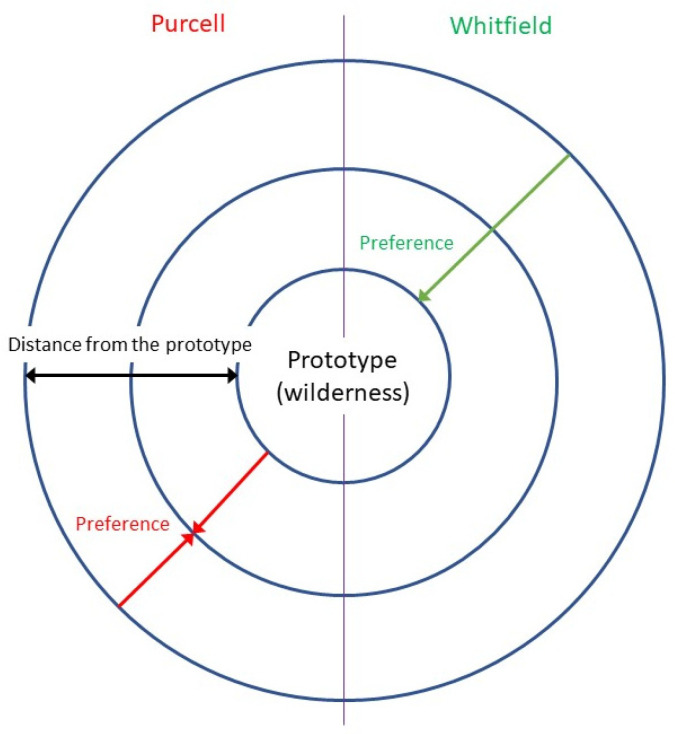
A comparison of the two models of preference: Purcell versus Whitfield. Source [59], modified by us.

**Table 1 ijerph-20-06354-t001:** Our working hypothesis: individuals perceive restorative and prefer the kind of Nature that matches their feeling of connection to Nature (source [14], modified by us).

Perceived Restoration	Preference for Typologies of Nature	Connection to Nature
High	Wilderness	High
Medium	Domesticated	Medium
Low	Urban	Low

**Table 2 ijerph-20-06354-t002:** Percentage of participants for the three levels of connection to Nature and the corresponding mean score (standard deviation in parenthesis).

	Percentage of Cases	CN Mean Score (SD)
Low	19.6%	2.30 (0.40)
Medium	32.2%	3.44 (0.34)
High	48.1%	4.59 (0.37)

**Table 3 ijerph-20-06354-t003:** Mean score (and SD in parenthesis) of the participants’ interpersonal (INTERP) and naturalistic (NATUR) intelligence across the three levels of connection to Nature.

CN	INTERP	NATUR
Low	2.80 (0.73)	4.14 (0.55)
Medium	2.98 (0.65)	4.49 (0.46)
High	3.25 (0.71)	4.75 (0.36)

**Table 4 ijerph-20-06354-t004:** Mean score (and standard deviation in parenthesis) of participants’ perceived restoration (PR), preference (PREF) and familiarity (FAM) for the three types of Nature and across the three levels of connection to Nature.

	CN	PR	PREF	FAM
**Wild**	Low	6.15 (2.20)	6.57 (2.40)	5.78 (2.96)
	Medium	7.25 (1.77)	7.82 (1.88)	6.74 (2.41)
	High	8.20 (1.58)	8.48 (1.69)	7.77 (2.17)
**Domesticated**	Low	6.31 (1.97)	6.91 (2.20)	4.96 (2.54)
	Medium	6.51 (2.12)	7.19 (2.47)	4.72 (2.70)
	High	7.05 (2.00)	7.66 (1.99)	5.92 (2.65)
**Urban**	Low	3.88 (2.39)	4.49 (2.80)	5.21 (3.38)
	Medium	3.69 (2.15)	4.37 (2.63)	5.35 (3.23)
	High	3.38 (2.09)	4.01 (2.70)	4.82 (3.29)

**Table 5 ijerph-20-06354-t005:** Pearson’s correlation between interpersonal (INTERP) and naturalistic (NATUR) intelligence, connection to Nature (CN), perceived restoration (PR), preference (PREF) and familiarity (FAM) for all participants.

	INTERP	NATUR	CN	PR
NATUR	0.22 **			
CN	0.25 **	0.52 **		
PR		0.18 **	0.23 **	
PREF		0.14 *	0.19 **	0.85 **
FAM			0.20 **	0.62 **

* *p* < 0.05; ** *p* < 0.01.

**Table 6 ijerph-20-06354-t006:** Pearson’s correlation between naturalistic (NATUR), connection to Nature (CN), perceived restoration (PR), preference (PREF) and familiarity (FAM) for the three levels of participants’ connection of Nature.

		NATUR	PR	PREF
Low CN	PR	0.26 *		
	PREF	0.25 *	0.78 **	
	FAM		0.57 **	0.57 **
Medium CN	CN	0.36 **		
	PREF		0.90 **	
	FAM		0.63 **	0.61 **
High CN				
	PREF		0.84 **	
	FAM		0.62 **	0.64 **

* *p* < 0.05; ** *p* < 0.01.

## Data Availability

The data presented in this study are available from the corresponding author upon reasonable request.

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
