# Peer review of "Wilderness Is the Prototype of Nature Regardless of the Individual’s Connection to Nature. An Empirical Verification of the Solastalgia Effect"

_ijerph, 2023, doi:10.3390/ijerph20146354_

Round 1
Reviewer 1 Report (Previous Reviewer 1)
Thank you for making the correction. I have reviewed the revised manuscript and found that most of my previous comments have been appropriately addressed. However, I noticed that the following two issues still require correction.
1. The parentheses in lines 325 and 329 should be corrected from (W, R, U) to (W, D, U).
2. It seems that there is a misunderstanding regarding Point 8. The corresponding paragraph can be located starting at line 329 in the revised manuscript. This paragraph discusses the results presented in Figure 2 (Table 4) and not Table 6, as mentioned in line 339. As I noted earlier, for instance, in line 311, the phrase "between PR*CN: F(4, 622) = 11.57, 331 p = .000" needs to be replaced with "interaction between types of Nature * levels of CN on PR: F(4, 622) = 11.57, 331 p = .000."
Author Response
Reviewer 1: Thank you for making the correction. I have reviewed the revised manuscript and found that most of my previous comments have been appropriately addressed. However, I noticed that the following two issues still require correction.
Response: We thank Reviewer 1 for suggestions.
Point 1. The parentheses in lines 325 and 329 should be corrected from (W, R, U) to (W, D, U).
Response 1: We corrected the errors in the parentheses in lines 325 and 329.
Point 2. It seems that there is a misunderstanding regarding Point 8. The corresponding paragraph can be located starting at line 329 in the revised manuscript. This paragraph discusses the results presented in Figure 2 (Table 4) and not Table 6, as mentioned in line 339. As I noted earlier, for instance, in line 311, the phrase "between PR*CN: F(4, 622) = 11.57, 331 p = .000" needs to be replaced with "interaction between types of Nature * levels of CN on PR: F(4, 622) = 11.57, 331 p = .000."
Response 2: We rephrased the sentences in lines 331-332 and in lines 339-340 as suggested by the Reviewer 1.
The revised paper is in attachment.

This manuscript is a resubmission of an earlier submission. The following is a list of the peer review reports and author responses from that submission.
Round 1
Reviewer 1 Report
I find the perspective and results of this study, which examines the correlation between one's affinity for nature and their evaluation of various natural environments, quite intriguing. Nevertheless, there were a few aspects that gave me cause for concern, and I would appreciate seeing them addressed.
l. 173
I see the rationale behind using natural intelligence as a variable in the study, but could you please clarify the reason for including interpersonal intelligence as well? I would appreciate a bit more elaboration on this point.
l. 177
Could you explain how the number of participants was determined for this study? Additionally, given that it's an online survey, is the sample size adequate or possibly too small? I'm particularly concerned about the Low CN group, which consists of only around 60 participants, and its implications for interpreting the results.
Page 6
When it comes to the three images that were used, do you believe they accurately represent wilderness (W), rural (R), and urban (U) environments? While I understand that they do highlight the variation in the amount of human intervention, I do have some reservations. For instance, the U image appears to have very little greenery, and the R image seems more like a well-manicured green space in an urban setting. Additionally, in my opinion, the W image looks more like a rural landscape rather than true wilderness.
l. 262
While it's been stated that the order of scales was randomized, I'm curious to know if the order in which the participants evaluated wilderness (W), rural (R), and urban (U) environments was also randomized.
Results section
With regards to the statistical analysis of the results, I noticed that only one degree of freedom for F was provided for each ANOVA outcome. It's important to also provide the degrees of freedom for residuals.
l. 266 and Table 2
I have a couple of questions about Table 2. First, could you please clarify the cutoff values used to define Low, Mid, and High CN? Additionally, I noticed that the notation for connection to nature is sometimes given as CN and sometimes as CNS, both in the tables and in the text. Could you please ensure that the notation is consistent throughout?
l. 274
Could you please explain the rationale for using LSD tests for post-hoc analysis? I'm asking because LSD tests are not ideal for controlling type I errors, and it's generally recommended to use alternative methods or adjust p-values using methods such as the Bonferroni correction.
l. 289
I noticed that the descriptions of the results in Table 4 include expressions like PR*CNS and FAM*CNS. However, these are not accurate because PR and FAM should be dependent variables rather than factors, and such interactions do not exist. Instead, these should refer to the CNS*WRU interactions on PR and FAM
Figure 2
Given that the scale scores range from 0 to 10, it's recommended that the y-axis of the graphs used to present the results should also span from 0 to 10. Additionally, as Table 4 and Figure 2 contain the same information in different formats, it would be more appropriate to present only one of them. Furthermore, when presenting results in graphical format, it's essential to include the mean values as well as the standard deviations, standard errors, or confidence intervals to provide a more comprehensive view of the data.
l. 318
I believe that referring to r = 0.52 as "quite good" may be an overstatement.
The title and description of Table 6 do not seem to correspond with its contents. The title mentions INTERP, but it is not included in the table. Moreover, I am confused about why the values for Low, Mid, and High CN in the first row of each group are different. Overall, this table is perplexing.​​​​​​​
Author Response
We thank Reviewer 1 for appreciating our study and for suggesting some important improvements. First of all the the intensity of human's intervention as a criterion for distinguishing the three types of Nature. Consequently we have replaced the term "rural" with "domesticated". Our responses point by point are in the attached file.

Reviewer 2 Report
In this potentially interesting study Authors have attempted to explore the relationship between preference of 3 types of nature (urban, rural and wild) with the perceived restorativeness and nature connectedness. The undertaken task of marrying various concepts is very ambitious and authors might need to work a bit more on the conceptual part of their manuscript.
- the difference between restorativeness and perceived restorativeness should be explicitly highlighted in the discussion, and study implications of the study.
- Authors seem to suggest that natural habitat assigned to a place of residence of an individual (familiar to them?) is perceived as the most restorative. It would be interesting to discuss how the level of familiarity moderated the preference of wilderness.
- The recruitment strategy, sampling method and procedures are missing. Were the participants blinded to the hypothesis (this may explain some contradictory results)? What was the dropout rate? Confounding factors?
Lines 141-153 and on - Authors' statements seem to base themselves on incomplete premise. The topic of types and components of different kinds of wilderness Nature (as authors define it) and their influence on mental health has been studied . Laboratory-based empirical studies for example on contemplative features of scenes and brain responses (including the restorativeness) to them were also conducted. Following this logic, the importance of quality of nature is not only well established, but also studied, and it looks like it is not that "the wilder the better", but there are some features and aspects of nature that humans respond to collectively, aside from individual preferences. Authors should better adjust their reasoning to the existing evidence and most recent literature.
Line 128 on - Why do authors link the lack of contact with nature with solastalgia, while concepts more commonly used in the field, such as nature deficit disorder are not discussed at all? The reasoning for that choice should be more pronounced, and not including other constructs (nature deficit disorder, mental fatigue, cognitive overload, overstimulation, urban stress etc) should be discussed.
- What are the practical implications and takeaways from this study?
Especially the discussion section should be improved in terms of English and style, in order to be more organized and comprehensive.
Author Response
We thank Reviewer 2 for appreciating our study. Reviewer 2's comments encouraged greater clarity in our thinking and accuracy in our manuscript. Our responses point by point are in the attached file.

Round 2
Reviewer 2 Report
The manuscript significantly improved after applying all the comments. I can recommend for publication.
The quality of English language is satisfactory.